

**Deriving Photosynthetically Active Radiation at ground level in**
**cloud-free conditions from Copernicus Atmospheric Monitoring**
**Service (CAMS) products**
**William Wandji Nyamsi[1], Philippe Blanc[2], John A. Augustine[3], Antti Arola[1] and Lucien**
**Wald[2]**
[1]{Finnish Meteorological Institute, Kuopio, Finland}
[2]{MINES ParisTech, PSL Research University, Centre Observation, Impacts, Energy,
Sophia Antipolis, France}
[3]{NOAA Earth System Research Laboratory, Global Monitoring Division (GMD), Boulder,
CO 80305}
Correspondence to: William Wandji Nyamsi (william.wandji@fmi.fi)
**Abstract**
A method is described that estimates the photosynthetically active radiation (PAR) at ground
level in cloud-free conditions. It uses a fast approximation of the libRadtran radiative transfer
numerical model, known as the *k*-distribution method and the correlated-*k* approximation of
Kato et al. (1999). LibRadtran provides irradiances aggregated over several fixed spectral
bands and a spectral resampling is proposed followed by an aggregation in the range [400,
700] nm. The Copernicus Atmosphere Monitoring Service (CAMS) produces daily estimates
of the aerosol properties, and total column contents in water vapor and ozone that are input to
the method. A comparison of the results is performed against instantaneous measurements of
global Photosynthetic Photon Flux Density (PPFD) on a horizontal plane made in cloud-free
conditions at seven sites of the Surface Radiation network (SURFRAD) in the USA in various
climates. The bias ranges between -12 µmol m$^{-2}$ s$^{-1}$ (-1% of the mean value at Desert Rock)
and +61 µmol m$^{-2}$ s$^{-1}$ (+5% at Penn. State Univ). The root mean square error ranges from
37 µmol m$^{-2}$ s$^{-1}$ (3%) to 82 µmol m$^{-2}$ s$^{-1}$ (6%). The coefficient of determination R$^2$ ranges
between 0.97 and 0.99. This work demonstrates the quality of the proposed method combined
with the CAMS products.





## 1. Introduction


Plants, algae, and certain microorganisms need solar radiation for their growth through the
photosynthesis process. The essential part of solar radiation to perform the photosynthesis is
that in the spectral band between 400 nm and 700 nm, and is called photosynthetically active
radiation (PAR). It is defined as the incident power per unit surface for this spectral interval
and can be expressed in W m$^{-2}$. PAR is also a measure of the photosynthetic photon flux density
(PPFD) and expressed in µmol m$^{-2}$ s$^{-1}$, and is defined as the number of the incident photons per
unit time per unit surface. Both units are linked by the widely used approximation
1 W m$^{-2}$ ≈ 4.57 µmol m$^{-2}$ s$^{-1}$ (McCree, 1972).
Specialists in agriculture over the world are in need of accurate estimates of amount of PAR
reaching the ground, including the assessment of the direct and diffuse components because of
their different influences on the plants. For example, diffuse light creates a more homogeneous
light profile in the canopy than direct light (Li et al., 2015). The sum of direct and diffuse PAR
is defined as the global PAR.
The use of appropriate instruments such as quantum sensor represents one way to provide and
also to respond to the increasing demand of PAR information. But PAR measurements are
spatio–temporally sparse due to instrument, maintenance and operation costs. This paucity
leads scientists to use broadband radiation as a proxy of PAR information because broadband
radiation measurements are more often available over time and in space. For instance, a
constant proportion (i.e. 2.079 µmol J$^{-1}$) of the daily mean of global broadband irradiance was
suggested by Udo and Aro (1999) to estimate the daily mean of global PAR. A proportion of
1.919 was proposed by Jacovides et al. (2004). These researchers recognize that the realistic
ratio depends on atmospheric conditions. There is a practical advantage to this approach as
broadband radiation is accurately estimable from satellite images (Blanc et al., 2011; Lefèvre
et al., 2014). Alternate sources of data on broadband radiation are meteorological analyses or
forecasts though their quality is poorer than that of satellite-derived data (Boilley and Wald,
2015; Bengulescu et al., 2017; Trolliet et al., 2017).
. Radiative transfer models (RTM) like libRadtran (Emde et al., 2016; Mayer and Kylling,
2005) are an alternative if accurate inputs are available that describe the atmosphere in cloud-
free conditions and the properties of the ground. RTMs are usually computationally expensive.
The *k*-distribution method and correlated-*k* approximation of Kato et al. (1999) is one strategy



adopted by libRadtran to reduce the amount of calculations and the computing time to obtain
the total solar irradiance. In this strategy, the computation of the irradiance is made in 32
spectral bands only in the range [240, 4606] nm and the results are then aggregated to yield the
total irradiance. Therefore, in the following, we will call these 32 spectral intervals as Kato
bands (KB). The band number will be in subscript. The operational McClear model is one
example of the use of this strategy (Lefèvre et al.; 2013). It estimates the total irradiance in
cloud-free conditions by making use of several abaci or look-up tables pre-computed with
libRadtran.
For each of the 32 KBs, a comparison of transmissivities computed by the Kato et al. approach
and summed from spectral detailed calculations over KB under concern for a set of 200,000
realistic cloud-free and cloudy atmospheres was carried out by Wandji Nyamsi et al. (2014).
They concluded that Kato et al. estimates are accurate and useful for representing irradiances
in each of the twelve KBs covering the PAR band.
The KBs do not cover correctly the PAR range. A spectral resampling technique has been
developed by Wandji Nyamsi et al. (2015) to overcome this difficulty. The concept of the
technique is to determine one or more narrow spectral bands of 1 nm width within each KB,
whose atmospheric transmissivities are correlated to that of the KB whatever the cloud-free
conditions. Eventually, the detailed 1 nm transmissivities over the range [400, 700] nm are
obtained by a linear interpolation process applied to these selected narrow bands and then
aggregated to yield the PAR irradiance. The technique has been validated against PAR
simulated by libRadtran and has revealed a very high accuracy for the global PAR and its direct
and diffuse components.
Now, the concept is tested for measurements of PAR fluxes operated at seven stations in the
USA limited to cloud–free conditions. The method combines the resampling technique and
atmospheric parameters as inputs. These latter are aerosol properties, total column ozone
(TOC) and total water vapor (TW) provided by the Copernicus Atmosphere Monitoring
Service (CAMS) at any place and time after 2003. This research represents a partition of a large
project aiming at producing an operational tool for assessing PAR taking benefit of the
availability of CAMS products.





## 2. Ground–based measurements used

PAR measurements were collected from seven locations of the Surface Radiation network (SURFRAD) in the USA (Figure 1). It is a well–known network established in order to support climate research with accurate, continuous, long-term measurements of the surface radiation budget over the USA (Augustine et al., 2000). The geographical coordinates and the code are given in Table 1. The LI–COR quantum sensor model LI–190 is currently used at all seven stations to provide measurements of the global PPFD received on a horizontal plane. No measurement of the direct or diffuse PAR is available. The measurements of high quality control are freely available and downloadable from the website ftp://aftp.cmdl.noaa.gov/data/radiation/surfrad/. The time period used for the validation is from 2010-01-01 to 2016-12-31, i.e. seven full years of measurements. Data are available as 1 min averages of 1 s samples. Calibration drift of the Quantum sensor the measures PAR in SURFRAD is checked in two operational practices. First, a general quality assurance measure is to replace monitoring radiometers annually with freshly calibrated units. That procedure discounts calibration drift over a period of years. Second, within a year, degradation of the PAR measurement is monitored in a routine daily "eye check" data quality control, as recommended by the Baseline Surface Radiation Network, or BSRN, (Ohmura et al., 1998). Each day the daily time series of the conversion factor (ratio of PAR to global broadband irradiance) is monitored. According to Pinker and Laszlo (1992), that ratio can vary between 0.4 and 0.65, depending on the solar zenithal angle, water vapor content, ozone, aerosols, and clouds. If the conversion factor falls below 0.4 and continues to decline, the instrument is replaced and the PAR data are corrected from the point when the drift began. The Bondville station is approximately 16 km southwest of Champaign, Illinois. It is located in a flat agricultural region with grasses and few trees. The Fort Peck station offers the same type of ground and is situated approximately 24 km north of Poplar, in northeastern Montana. The snow cover in Fort Peck presents a high interannual variation. The Penn. State Univ. station is located in a wide Appalachian valley on an agricultural research farm approximately 10 km southwest of State College, Pennsylvania. The surroundings are about three–fourths grass and one–fourth crops (southwest corner). The Goodwin Creek station is located approximately 32 km west of Oxford, Mississippi in a rural pasture while the Sioux Falls station is located on the grounds of the EROS Data Center, outside Sioux Falls, South Dakota.



The Table Mountain station is approximately 13 km north of Boulder, Colorado, and a few
kilometers east of the foothills of the Rocky Mountains. The surface is sandy with a mix of
exposed rocks, sparse grasses, desert shrubs, and small cactus. The character of the underlying
flora there changes seasonally, that is, it is usually green in the late spring and early summer,
and browns significantly by midsummer. Also in desert type landscape, the Desert Rock station
is located approximately 105 km northwest of Las Vegas, Nevada. It experiences a hot arid
climate. The ground is mostly made of fine rock and scattered desert shrubs. Practically, it does
not have any seasonal change of vegetation.
The accuracy of all quantum sensors is ±5% (Augustine et al., 2000). From the manufacturer
of    LI-COR    instrument    manufacturer,    the    total    error    is    about    8%
(https://www.licor.com/documents/3bjwy50xsb49jqof0wz4). In addition, at each station, the
diffuse, and global broadband irradiances at ground level, the direct normal irradiance, as well
as atmospheric pressure are measured every 1 min. Here, broadband means the interval from
280 nm to 2800 nm. Assuming that any cloud-free instant in PAR can be detected by detecting
cloud-free instants in a series of broadband irradiances, the Lefèvre et al. (2013) algorithm has
been applied to these three series of irradiances yielding a series of cloud-free instants
instances. Two consecutive filters composes the algorithm. The first one is a constraint on the
amount of diffuse broadband irradiance so that the diffuse is always lower than the global
broadband irradiance with a ratio less than 0.3 according that direct irradiance is obviously
dominant in cloud-free conditions. The second filter inspects the temporal variability of the
global broadband irradiance normalized by the broadband irradiance received at the top of the
atmosphere and by a typical air mass since this quantity should be steady for several hours in
cloud-free conditions. We assume that a cloud-free instant detected by analyzing broadband
irradiances is also cloud-free for the PAR measurements. It is possible that PAR is affected in
certain cases by the presence of scattered cloudiness which may go unnoticed in the broadband
range and that the retained series of cloud-free instants for broadband may comprise cloudy
instants for PAR. Given the large contribution of the PAR irradiance to the broadband
irradiance, and the high selectivity of the algorithm of Lefèvre et al. (2013), we believe that
such cases are rare and that the conclusions will be unaffected as a whole. Table 1 also give
the number of cloud-free instants.



## 3. The method

Briefly written, the method performs a spectral resampling every 1 nm of the transmissivities obtained by libRadtran in the twelve KBs spanning the PAR range and aggregates the resulting fluxes in the range [400, 700] nm.

### 3.1. Inputs to libRadtran

The PAR depends mostly on the solar zenithal angle $\Theta_s$, the ground albedo, the TOC and TWV, the vertical profiles of temperature, pressure, density, and volume mixing ratio for gases as a function of altitude, the aerosol optical depth (AOD) and type, and the elevation of the ground above sea level in cloud-free conditions. The origins of inputs are selected with the respect that the method shall be used operationally to provide estimates of PAR –irradiance and PPFD– at any location and any time The McClear model offers such kind of inputs. We have adopted the origins of inputs used by McClear (Lefèvre et al., 2013). In short, TOC and TWV and aerosol optical depths for black carbon, dust, organic matter, sea salt, and sulfate originate from CAMS. The SG2 algorithm gives $\Theta_s$ for the sun position and angle. (Blanc and Wald, 2012). The vertical profiles are taken from the AFGL data sets and a map indicates which one to use at any location (Lefèvre et al., 2013). The Shuttle Radar Topography Mission, source for ground elevation, provides the digital terrain model. As for the albedo, Bosch et al. (2009) proposed as a first approximation, to estimate the albedo in the PAR range by multiplying the broadband albedo by 0.47 if no information on the type of surface is available. The albedo is defined as the ratio of reflected to incident flux in a given spectral band at the surface. It is also defined the integral of the bidirectional reflectance distribution function (BRDF), depending on the surface-type and its roughness. We have used a series of maps of the MODIS-derived BRDF parameters for each calendar month for the broadband albedo with no missing values at a spatial resolution of 0.05° proposed by Blanc et al. (2014).

Actually for the sake of the simplicity, access to the inputs is made by automatic calls to the Web service McClear on the SoDa Service (Gschwind et al., 2016, www.soda-pro.com). In the verbose mode, the flow returned by McClear contains 1 min values of the inputs listed above. This can be conveniently exploited for the comparison with ground measurements.



### 3.2.The resampling technique and the new method

Wandji Nyamsi et al. (2015) has presented the concept of the resampling technique for the PAR range and Wandji Nyamsi et al. (2017) for the UV range. Regarding the PAR, Wandji Nyamsi et al. (2015) have found that for each $KB_j$ of the twelve KBs encompassing the PAR range, there are one or more 1-nm spectral intervals, denoted $NB_i$, whose transmissivities are correlated to that of the $KB_j$ by the means of affine functions. A total of 19 $NB_i$ is sufficient; the slope and intercepts of affine functions are reported in Table 2. The choice of these $NB_i$ has been made on an empirical basis.

The method is as follows. A run by libRadtran provides the fluxes in each of the 12 KBs. Then, one obtains the fluxes at each of the 19 $NB_i$ by using the functions in Table 2. A simple linear interpolation technique is applied to these 19 known fluxes to compute the fluxes every 1 nm in the PAR range [400, 700] nm. Eventually, the 1 nm fluxes are summed up to yield the PAR.

Wandji Nyamsi et al. (2015) made a numerical validation by comparing the results obtained by the new method to those given by libRadtran. For the direct and global PAR fluxes, the absolute values of the relative biases and the relative root mean square errors were less than 1%. They concluded that the new method performs very well for PAR estimates, both direct and global, and is much less demanding in computer resource and time than spectrally-detailed runs of libRadtran.

### 4. Results

The results of the proposed method were compared to 1 min measurements of global Photosynthetic Photon Flux Density for cloud-free conditions. The differences (estimation minus measurement) between the results of the method and the measurements were computed for each instant and the statistical indicators for measuring the performance of the method were then computed. They were the bias (mean of the errors), the root mean square error (RMSE), and their values *rbias* and *rRMSE* relative to the mean value of the measurements as well as the coefficient of determination ($R^2$). An analysis of the dependence of results with the month and the year was also carried out.

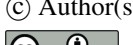



Figure 2 displays the 2D histogram, also called scatter density plot, of ground-based
instantaneous measurements and estimates at Fort Peck in cloud-free conditions. The cloud of
points is lengthened along the 1:1 line. The slope of the fitting line is 1.03, i.e. very close to 1,
showing a very good estimation of the measurements by the method. Estimates and
measurements are very well correlated with $R^2$ equal to 0.98. In addition, this high value means
that all the variability in the measurements is very well explained by the estimates. The bias is
low: +11 µmol m$^{-2}$ s$^{-1}$, i.e. +1% of the mean value of the measurements: 1262 µmol m$^{-2}$ s$^{-1}$.
The RMSE is small: 58 µmol m$^{-2}$ s$^{-1}$, around 5% of the mean.
Two red lines are plotted in the scatter density plot and correspond to relative differences within
±10%. One observes that the majority of points are within the two red lines. A set of points
marked with a red circle is seen where the method underestimates noticeably by more than
20%. These underestimations occur during a single day, 2$^{nd}$ July 2015. During that day, the
AOD at 500 nm from CAMS was 1.80 in average while the AOD measured at the station was
0.14 i.e. a factor ~10–12 of the measured value. As the greater the aerosol load, the smaller the
direct component of PPFD, and since the direct component is the major contributor to the
global PPFD in cloud-free conditions, the differences between the CAMS and measured AOD
mainly explain these underestimations. The relative errors vary slightly from one year to
another year.
The dependence of errors with variables was investigated. Figure 3 displays the ratio estimated
/ measured (top) and difference (estimated – measured, bottom) as function of $\Theta_s$, albedo, TOC,
TWV, and AOD at 550 nm. In general, there is no clear dependence of errors with variable
except for AOD where the errors show a tendency to decrease with increasing AOD. For the
ratio as well as the difference, the limits of boxes are close from one quartile to another meaning
a very limited spread. The deviations between maximum and minimum are small. The median
is similar to the mean. Whatever the number of data (in pink color), they are very close to 1
except for high AOD.
Results for the other stations are shown in Figure 4. The statistical indicators are reported in
Table 2. In general, the points lie along the identity line with a high density of points within
±10% of relative differences. $R^2$ is always greater than 0.97 for any station, meaning that
variability in global PPFD is very well reproduced by the estimates. In general, the points lie
along the identity line with a high density of points within ±10% of relative differences. The



method overestimates in all stations except Desert Rock. The bias varies
between -12 µmol m$^{-2}$ s$^{-1}$ (Desert Rock, relative bias of -1%) and 82 µmol m$^{-2}$ s$^{-1}$ (Penn. State.
Univ., relative bias of 5%).
The ground at Penn. State. Univ is most of time covered by grass and crop. In such cases, the
mean ratio recommended by Bosch et al., (2009) should be close to 0.2–0.3 instead of 0.47 as
assumed in the proposed method, yielding a smaller PAR albedo. The smaller the PAR albedo,
the smaller the contribution of the ground to the diffuse PAR, and the smaller the global PAR.
As a result, using a smaller PAR albedo would likely yield a smaller bias.
The performance of the method does not show a clear dependence with a specific month and
year. The dependency of errors with variable was also investigated. The maximum of ratios or
differences is lower for all stations than the one for Fort Peck because of this huge difference
in AOD on 2$^{nd}$ July 2015. For SZA, TOC, albedo and AOD, results similar results to those for
Fort Peck were found for the six other stations. Figure 5 shows the ratio as a function of TWV.
In general, the errors, both ratio and difference show a tendency to slightly increase with
increasing TWV except Desert Rock.
The absolute values of the bias (not shown) decrease with increasing $\Theta_s$ except Desert Rock.
One observes that the relative bias (not shown) increase with $\Theta_s$. Depending on the station, the
relative bias as function of $\Theta_s$ varies from positive values to negative values. For all stations
and whatever $\Theta_s$, the relative bias (not shown) and RMSE (not shown) are within ±6%. This
means a very limited scattering for all $\Theta_s$ and shows the high accuracy of the proposed method
whatever $\Theta_s$.

**5.  Conclusions**
In this work, we proposed a new method for estimating the PAR fluxes. It exploits CAMS
products as inputs. In addition, a spectral resampling technique is applied on the results of the
*k*-distribution method and the correlated-*k* approximation of Kato et al. (1999) followed by an
aggregation to provide PAR fluxes. The method has been validated on global PAR fluxes by
comparing its estimates to 1 min global PAR for seven stations located in USA in various
climates. In all stations, the coefficient of determination is greater than 0.97 denoting that a



very large part of variability in the measurements is captured by the estimates from the
proposed method. The relative bias ranges between –1% and +5% of the mean value of
measurements. The relative RMSE is very close to relative bias indicating a very small standard
deviation. These errors are less than 5% close to, or even less in most of cases, than the
uncertainty of the measurements. This demonstrates the very good level of accuracy of the
proposed method.
Despite the fact that the validations was carried out only on global PAR fluxes, the method is
also very useful for accurate direct and diffuse PAR estimates. The PAR estimates in KBs
could be operated quite fast by taking benefits of pre-computed abaci made for McClear model
which is $10^5$ times faster but still accurate approximation of results from forward radiative
transfer modelling with libRadtran.
Regarding that the proposed method offers accurate PAR estimates in cloud-free conditions,
one advantage of such kind of method is that any approach taking into account the attenuation
due to clouds could be applied on the method to provide all–sky PAR estimates. An example
of approach may be the use of the cloud modification factor as developed by Oumbe et al.
(2014) or similar approximations made by Huang et al. (2011) for total irradiance, or Calbo et
al (2005), den Outer et al. (2010) or Krotkov et al. (2001) for UV. The use of more accurate
surface albedo in the PAR range as input to the method following a approach similar to what
was suggested by Wandji Nyamsi et al. (2017) for UV, could improve PAR estimates in cloud–
free and all–sky conditions.

**6.  Data availability**
All PAR measurements at each station were provided by the SURFRAD network established
in 1993 through the support of NOAA's Office of Global Programs. High quality measurements
used          here          are          freely          available          and          were          downloaded          from
ftp://aftp.cmdl.noaa.gov/data/radiation/surfrad/
Products from CAMS are freely available after registration and were downloaded from:
http://atmosphere.copernicus.eu/





The McClear products are freely available after registration and were downloaded from: http://www.soda-pro.com

The BRDF maps by Blanc et al. (2014) may be downloaded from: http://www.oie.mines-paristech.fr/Valorisation/Outils/AlbedoSol/.

*Acknowledgments*. The authors thank NOAA ESRL Global Monitoring Division, Boulder, Colorado, USA (http://esrl.noaa.gov/gmd/) to make freely accessible SURFRAD data.

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





**Table 1**: Description of stations used for validation, ordered by decreasing latitude

| Station | Fort Peck | Sioux Falls | Penn. Sate Univ | Table Mountain | Bondville | Desert Rock | Goodwin Creek |
|---|---|---|---|---|---|---|---|
| Code | FPK | SXF | PSU | TBL | BND | DRA | GCM |
| Latitude (°) | 48.31 | 43.73 | 40.72 | 40.12 | 40.05 | 36.62 | 34.25 |
| Longitude (°) | −105.10 | −96.62 | −77.93 | −105.24 | −88.37 | −116.02 | −89.87 |
| Altitude (m) | 634 | 473 | 376 | 1689 | 230 | 1007 | 98 |
| NCFI* | 186698 | 245355 | 120097 | 260509 | 196871 | 603727 | 230420 |

*NCFI: Number of cloud-free instants.



Table 2. KB covering the PAR range and selected sub-intervals $NB_i$, slopes and intercepts of
the affine functions between the transmittances in KB and 1 nm $NB_i$.

| KB | Interval $\Delta\lambda$, nm | Sub-interval $NB_i$, nm (# i) | Direct normal | | Global | |
|---|---|---|---|---|---|---|
| | | | Slope | Intercept | Slope | Intercept |
| 6 | 363 – 408 | 385 – 386 (#1) | 0.9987 | -0.0023 | 1.0030 | -0.0032 |
| 7 | 408 – 452 | 430 – 431 (#2) | 1.0026 | -0.0004 | 0.9995 | 0.0013 |
| 8 | 452 – 518 | 484 – 485 (#3) | 1.0034 | 0.0005 | 0.9979 | 0.0000 |
| 9 | 518 – 540 | 528 – 529 (#4) | 0.9998 | -0.0005 | 1.0008 | -0.0013 |
| 10 | 540 – 550 | 545 – 546 (#5) | 1.0001 | 0.0003 | 1.0003 | -0.0003 |
| 11 | 550 – 567 | 558 – 559 (#6) | 1.0004 | 0.0004 | 0.9997 | 0.0012 |
| 12 | 567 – 605 | 569 – 570 (#7) | 0.9960 | -0.0119 | 1.0024 | -0.0100 |
| | | 586 – 587 (#8) | 1.0123 | 0.0064 | 0.9929 | 0.0267 |
| | | 589 – 590 (#9) | 0.9568 | -0.0109 | 0.9804 | -0.0434 |
| | | 602 – 603 (#10) | 1.0150 | 0.0167 | 1.0051 | 0.0212 |
| 13 | 605 – 625 | 615 – 616 (#11) | 1.0004 | 0.0009 | 0.9977 | 0.0033 |
| 14 | 625 – 667 | 625 – 626 (#12) | 1.0104 | -0.0174 | 1.0622 | -0.0551 |
| | | 644 – 645 (#13) | 1.0072 | 0.0029 | 0.9960 | 0.0154 |
| | | 656 – 657 (#14) | 0.9915 | 0.0068 | 0.9698 | 0.0205 |
| 15 | 667 – 684 | 675 – 676 (#15) | 1.0006 | 0.0007 | 0.9978 | 0.0036 |
| 16 | 684 – 704 | 685 – 686 (#16) | 1.0473 | 0.0212 | 0.9681 | 0.1036 |
| | | 687 – 688 (#17) | 0.9602 | -0.0130 | 1.0041 | -0.0531 |
| | | 694 – 695 (#18) | 0.9828 | -0.0153 | 1.0323 | -0.0642 |
| 17 | 704 – 743 | 715 – 716 (#19) | 1.0262 | 0.0121 | 0.9771 | 0.0596 |






Table 3. Statistical indicators of the performances of the method. Mean, bias and RMSE are
expressed in ($\mu$mol m$^{-2}$ s$^{-1}$). N is the number of samples.

| Station | N | Mean | Bias | RMSE | rBias (%) | rRMSE (%) | $R^2$ |
|---|---|---|---|---|---|---|---|
| Fort Peck | 186698 | 1262 | 11 | 58 | 1 | 5 | 0.98 |
| Sioux Falls | 245355 | 1247 | 1 | 53 | 0 | 4 | 0.98 |
| Penn. State Univ | 120097 | 1273 | 61 | 82 | 5 | 6 | 0.98 |
| Table Mountain | 260509 | 1263 | 50 | 69 | 4 | 5 | 0.99 |
| Bondville | 196871 | 1257 | 36 | 74 | 3 | 6 | 0.97 |
| Desert Rock | 603727 | 1424 | –12 | 37 | -1 | 3 | 0.99 |
| Goodwin Creek | 230420 | 1320 | 42 | 70 | 3 | 5 | 0.98 |








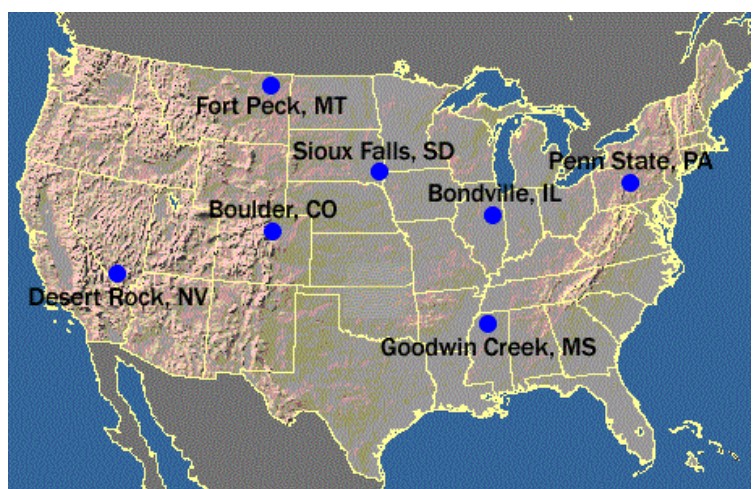

410 Figure 1: Map of SURFRAD stations (courtesy of NOAA).







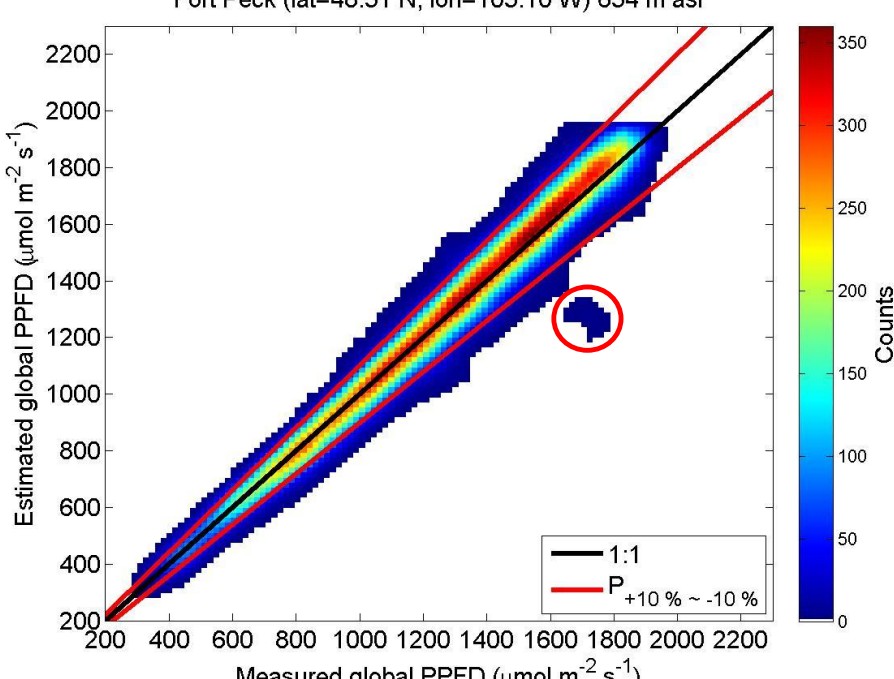


Figure 2: 2D histogram of 1 min measurements of global PPFD and estimates at Fort Peck. The bin width of the histogram is 20 $\mu$mol m$^{-2}$ s$^{-1}$ and the colorbar indicates the number of points in each bin.









Figure 3: Dependence of ratio (top) of the estimated (Esti) to the measured (Meas) and difference between estimated and measured (bottom) with each variable for Fort Peck. The variable is indicated at top of each plot. The red dots indicates the mean, the limits of the boxes are 1$^{st}$, 2$^{nd}$ (median), 3$^{rd}$ quartiles. The lower whisker is the minimum and the upper one is the maximum. The pink number is the number of data in a single variable range



Figure 4 Same as Fig. 2, but all stations except Fort Peck. The station name is indicated at top.

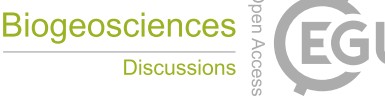



428

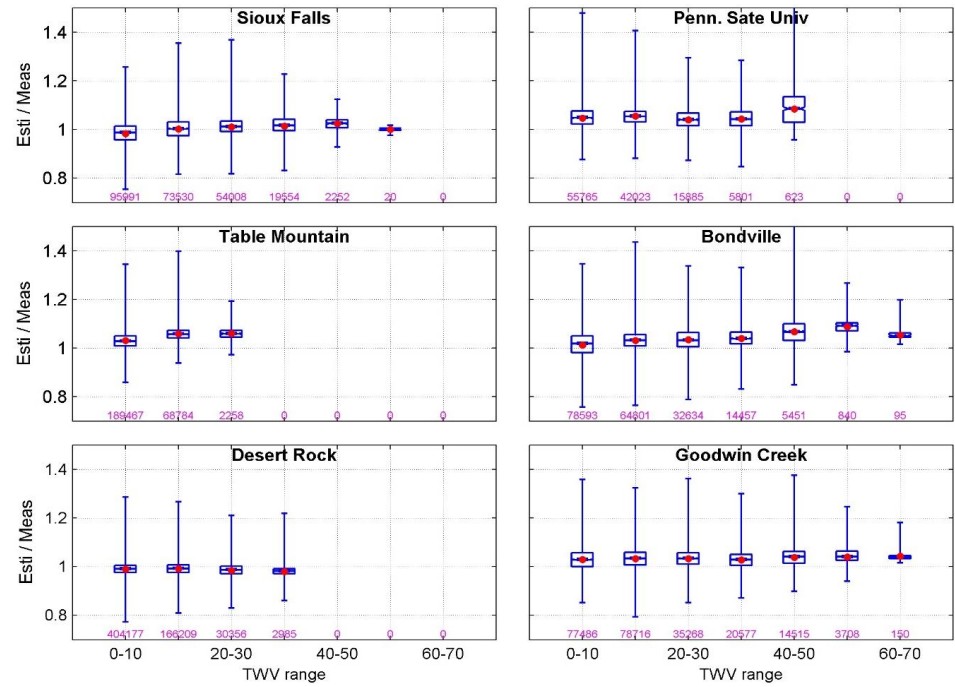

429

Figure 5: Dependence of ratio of the estimated (Esti) to the measured (Meas) with TWV
range for each station except Fort Peck.