# Peer review of "Deriving Photosynthetically Active Radiation at ground level in"

_Biogeosciences, 2017_

## Referee Comment (RC1) · Anonymous Referee #1 · 5 Feb 2018

General comments: The manuscript "Deriving Photosynthetically 1 Active Radiation at ground level in cloud-free conditions from Copernicus Atmospheric Monitoring Service (CAMS) products" describes a method of calculating the photosynthetically active radiation (PAR) during cloud-free conditions from widely available CAMS data using the libRadtran radiative transfer numerical model in combination with a spectral resampling procedure. Since PAR is an important variable to assess the carbon and water cycle of the terrestrial biosphere and is used in most process models and statistical models, this research paper and the research question described within is of high interest for

the research community. However, overall the manuscript suffers from a lack of novelty regarding the presented method. The actual new portion of the work is not much. While most of the method such as the spectral resampling has been developed and published already before, and the authors also refer to that appropriately. Yet, the new part is so little that it is in fact summarized in only 4 lines by the authors, in particular lines 190 to 193 in the manuscript. The rest is simple interpolation and regression analysis as well as an application of the libRadtran library functions in order to successfully relate PAR to other measurements which is not new at all even though, a specific dataset was used for the predictor variables. The new contribution seems rather minor for a publication in Biogeoscience and for that matter for a research paper. I suggest enhancing your study to non-cloudy conditions as alluded to at the end of the manuscript (lines 281 to 289), for instance. Since the amount of PAR at the surface is clearly related to the total irradiation, cloudiness and atmospheric variables such as aerosol optical depth and other meteorological conditions this seems feasible and would more of an advance in this field of research over previous studies. I therefore suggest a major review/overhaul of the manuscript.

Specific comments: 1. In line 44 to 54 the authors give only few references. Due to the importance of PAR and the sparsity of its measurements similar research has been going on for many years. Please include more studies that also aim to model PAR by using atmospheric measurements as predictor variables. Also, the performance of the approaches presented in those different studies should be compared to your findings to put your results into context of this ongoing research in his field.

2. In the paragraph starting in line 44 the authors also discuss the quality of the PAR measurements and also address the error of the PAR devices. This is a bit too short given the complexity of this issue. The error of the ground measurements, when used a reference, are of outstanding importance for the model presented. There are studies already published that address the errors of PAR measurements in practice at the ground level. Schmidt et al. (2012) ,for instance, give an average error of 3% for PAR

**BGD**

measurements among a large network that is, however, affected by a large standard deviation of errors observed between PAR sensors at different sites and a reference system. A PAR sensor's response to the natural spectrum needs to be addressed briefly. Although the authors describe the quality control of the measurements used, the fact that measurements might always show some deviance from the real value should be addressed on a certain level in a manuscript that focusses on PAR. The response of the very common PAR sensor (LICOR quantum sensor) that is also used in this study is not perfect for the natural spectrum but adheres to an optimized spectral curve. Please address this issue if the performance of your approach is measured by the correlation with those ground measurements.

3. The sentence in lines 75 to 78 is not quite clear, neither is the approach described. Although a reference is given, please rephrase and add a brief description about the method. How many narrow bands are used in your case? The authors mention "one or more". I would expect the model performs better the more band bins are correlated and used. How strong do the correlations have to be? The summary is too simple to understand the process. Later in section 3.2 the method is explained in more detail, yet the statement "A total of 19 NBi is sufficient" comes out of nowhere. Please give information as to why 19 can be considered sufficient. Please also merge these two sections and explain this part of the method only once but thoroughly.

Details about the origin of the function parameters (slope and intercept) are not given although it seems to be the central point of the method. How many comparisons at which stations were used to get these parameters of the affine functions? A statement like "The choice of these NBi has been made on an empirical basis." As give in lines 188 and 189 is not a sufficient description of the process. Please elaborate.

4. Figure 3 gives the differences between the measured (at the sites) and CAMS data. While the upper panel gives a ratio, the lower panel gives the absolute difference divided by 100. It is not clear why the difference given in the original data units should be divided by 100 (unless one wants to make them appear much small than they are).

Also, please provide unit labels on the vertical axes showing the differences of the various variables in the lower panel.

Technical corrections: The sentence in line 75 starts with a dot. Please correct (delete). In line 163 there is a dot missing at the end of the sentence. "...any location and any time." Line 167: Please spell out AFGL data set. The acronym appears for the first time here and not any reader might be familiar with that dataset and its origin. Line 137 and 138: The sentence reads "...cloud-free instants instances." Please correct.

Reference used in comments:

Schmidt A., C. Hanson, W.S. Chan, B.E. Law (2012): Empirical assessment of uncertainties of meteorological parameters and turbulent fluxes in the AmeriFlux network. Journal of Geophysical Research 117, G04014, doi:10.1029/2012JG002100.

---

## Referee Comment (RC2) · Anonymous Referee #2 · 9 Feb 2018

Background

In 1999 Nyamsi et al. published in Adv. Sci. Res., 12, 5-10, doi: 10.5194/ a description of an approach to derive PAR under clear sky conditions from the correlated-k approximation of Kato et al. (1999). This approach is computationally efficient, initially applied for calculations of the broadband solar radiation under clear sky conditions in 32 specific spectral bands. The authors used this approach for assessment of the photosynthetically active radiation (PAR) from 400 to 700 nm using twelve of these spectral bands. The method has been evaluated against detailed spectral calculations of PAR

derived with the radiative transfer model libRadtran.

In the present study, the Copernicus Atmosphere Monitoring Service daily estimates of aerosol properties, and column contents of water vapor and ozone are used as input to the method to derive PAR under clear sky condition. The results are compared with measurements of global Photosynthetic Photon Flux Density on a horizontal plane made in cloud-free conditions at seven sites of the Surface Radiation network (SURFRAD) in the USA.

General Comments

1. The methodology used here has been described previously. 2. The methodology has been implemented with actual auxiliary data to match ground observations of PAR. It was found that the bias ranges between 1-6 % from the mean value. It is claimed that these errors are less than 5% than the uncertainty of the measurements. It is claimed that this demonstrates the very good level of accuracy of the proposed method (which is not obvious how). 3. Not clear what is the added value of this evaluation since the methodology itself was already evaluated. Seems, this is just an exercise what is achieved if the CAMS product is used. Would it be worse with other sources of products? 4. Not clear how this work brings us closer to obtain information on PAR under all sky conditions (information that is needed). 5. The approach proposed is not unique and in principle, any radiative transfer model can be used to estimate PAR. Therefore, the unique contribution of the described effort was not demonstrated clearly and neither has it been shown how this gets us closer to obtain large scale information on PAR under all conditions. 6. Relevant references are very limited. 7. Addressing all of above concerns is needed before considering publication.

---

## Referee Comment (RC3) · Anonymous Referee #3 · 17 Feb 2018

The manuscript submitted to Biogeosciences titled, "Deriving Photosynthetically Active Radiation at ground level in cloud-free conditions from Copernicus Atmospheric Monitoring Service (CAMS) products" by Wandji Nyamsi et al. presents one aspect of a larger project developing radiation modeling for the bottom of the atmosphere. This includes a sequence of recent articles lead by the current lead author. The specific objective of this study appears to be testing the use of CAMS atmospheric products for inputs to an existing numerical radiation transfer model. The model itself is cited, as is the spectral resampling technique used here, referred to as the 'new method'. What

is new in THIS study seems to be the use of the CAMS products; columnar aerosol optical depth (AOD), ozone (O3) and water vapor (H2O) inputs to the radiation model.

The general accuracy of the simulated cloud-free PAR is impressively high at 7 locations with strong regional differences likely in AOD and H2O. However, a number of questions seem to remain that are central to the research objective. These include, • What causes the tendency of the model to perform well at lower PAR levels and over-estimate at higher values (e.g. Fig. 2), and cause slope >1 in general? Did this appear in previous evaluations of the model? • Can the differences in accuracy found between the seven stations be used to inform the cause of inaccuracy? • The model over-estimate appears to be correlated with a tendency of the model to overestimate under low zenith angles and/or under low AOD. Why is this? • Is the cause of this inaccuracy related to the CAMS data set or is it the radiation model itself?

The answers to these questions seem important to the objectives of this study, because they should provide leads towards future model improvements. As presented, I find the manuscript needs to overcome two major obstacles currently inhibiting its potentially useful contribution to Biogeosciences.

1. Surprisingly, little information is provided about the CAMS product, especially accuracy assessments of AOD, O3 and H20 vapor products from past studies. This context is necessary to assess the efficacy of using it for inputs to radiation modeling. It may also provide some useful insight into the cause of the over-estimate of PAR in simulations found here.

2. Analysis doesn't seem to test the CAMS input specifically, but rather the CAMS-driven radiation simulations against observations. This could be done by comparing these simulations to those using standard (e.g. monthly average) AOD, H20 and O3 for a site. Sensitivity simulations for variations in these input variables would also help shed light on the sensitivity of model accuracy to each of these inputs (or discussed, if reported elsewhere). My guess is that this shows AOD is key. I believe AOD is

also measured at the observation sites? Could the CAMS product be evaluated to determine if, for example, the over-estimate could be related to a bias in CAMS AOD?

Minor comments: Specific objective of this study is not really clear. Nor is it clearly separated from the other model developments in this sequence. There seems to be significant overlap as written.

It would be helpful to the readership of this journal for the authors to articulate more fully in the introduction and discussion sections, the implications of these methods and results to biogeoscience research in general.

Methods, why not include the Modeling Efficiency Coefficient (Nash and Sutcliff 1970)? It's a very direct test of model performance, including bias and random error.

There is no discussion section. This would be a useful place to investigate answers to the questions above based on the evidence presented in results.

The manuscript needs to be carefully edited for small but frequent lapses in grammar, or clarity in expression.

L128: What is 'fine rock'?

L137: End of sentence unclear - 'instant instances' is confusing alliteration. Anyway to rewrite using other words such as 'periods' , 'frequency'. . .?

In discussing Fig's 2 and 4, What is an 'identity line'?
* * *

---

## Author Comment (AC1) · 20 Apr 2018

First of all, we thank Referee #1 for these constructive remarks on this topic. The authors believe that they have understood the concerns of the referee. All remarks have been taken into account for revising a part of the text following recommendations of the referee.

General comments: The manuscript "Deriving Photosynthetically Active Radiation at ground level in cloud-free conditions from Copernicus Atmospheric Monitoring Service

(CAMS) products" describes a method of calculating the photosynthetically active radiation (PAR) during cloud-free conditions from widely available CAMS data using the libRadtran radiative transfer numerical model in combination with a spectral resampling procedure. Since PAR is an important variable to assess the carbon and water cycle of the terrestrial biosphere and is used in most process models and statistical models, this research paper and the research question described within is of high interest for the research community. However, overall the manuscript suffers from a lack of novelty regarding the presented method. The actual new portion of the work is not much. While most of the method such as the spectral resampling has been developed and published already before, and the authors also refer to that appropriately. Yet, the new part is so little that it is in fact summarized in only 4 lines by the authors, in particular lines190 to 193 in the manuscript. The rest is simple interpolation and regression analysis as well as an application of the libRadtran library functions in order to success fully relate PAR to other measurements which is not new at all even though, a specific dataset was used for the predictor variables. The new contribution seems rather minor for a publication in Biogeoscience and for that matter for a research paper. I suggest enhancing your study to non-cloudy conditions as alluded to at the end of the manuscript (lines 281 to 289), for instance. Since the amount of PAR at the surface is clearly related to the total irradiation, cloudiness and atmospheric variables such as aerosol optical depth and other meteorological conditions this seems feasible and would more of an advance in this field of research over previous studies. I therefore suggest a major review/overhaul of the manuscript

Answer: Thank you for this comment. We feel that apparently we were not able to emphasize clearly enough those parts of the method that have been already published from those that are discussed and published first time. We appreciate this comment and we have tried to clarify these issues in the revised manuscript. Moreover, we want to stress that the core objective was to validate this approach (at PAR range) against ground-based measurements. This is done first time in this manuscript and is entirely new contribution.

**BGD**
The method we described is a combination of three parts: (1) use of CAMS products to describe the atmospheric state, (2) irradiances of correlated–k approach over only eleven bands covering the PAR wavelengths by the means of libRadtran and (3) the resampling technique for computing PAR estimates. Only the third part has been previously published by ourselves. The goal of this current manuscript is to focus on the entire approach (e.g. also including the other two parts) and to present the ground-based validation.

Since estimation of PAR under cloud-free conditions at any time and place is an important first step in calculating PAR in all-sky conditions, in this paper, we concentrated first on these conditions.

Based on the above referee comment, we have re-written several parts of the text accordingly. For instance, a part of introduction is re-written as follows: "This resampling technique has not been validated in operational conditions, i.e. using available inputs to describe the atmosphere in cloud-free conditions and the properties of the ground, and tested against ground-based measurements. This paper is making this step forward and aims at describing and evaluating the entire method when tested against measured PAR in cloud-free conditions."

It is also important to emphasize that the second part of the method can be obtained by the means of other radiative transfer models using correlated–k approach such as Doubling Adding KNMI (DAK), Rapid Radiative (RAPRAD) transfer, SPECMAGIC. We have also stressed the universality of entire method when correlated–k estimates are available.

Specific comments:

Comment 1. In line 44 to 54 the authors give only few references. Due to the importance of PAR and the sparsity of its measurements similar research has been going on for many years. Please include more studies that also aim to model PAR by using atmospheric measurements as predictor variables. Also, the performance of the ap-

**BGD**
proaches presented in those different studies should be compared to your findings to put your results into context of this ongoing research in his field.

Answer: Thank you for this remark. We fully agree with you. We have re-written the relevant part of the paragraph and have added a paragraph in the text as follows:

"Therefore, several authors have developed methods for estimating PAR in cloud-free conditions by using different sources of atmospheric measurements as predictor variables, and the effect of clouds accounted for separately by an appropriate attenuation or modification factor (Oumbe et al., 2014). Su et al. (2007) has proposed such method with Clouds and the Earth's Radiant Energy System (CERES) products providing atmospheric conditions. Their method mostly shows a positive relative bias reaching up to 7% when validated with PAR measurements at seven Surface Radiation Budget Network (SURFRAD) sites in cloud-free conditions. Following the same idea, Bosch et al. (2008) has developed a parametrization for PAR estimates in cloud-free conditions. The relative bias was less than 1% when validated with three of the seven SURFRAD sites. It is important to mention that the authors used mainly ground-based atmospheric conditions. More recently, Sun et al. (2017) has proposed a method using solar radiation in the UV-visible spectral band. They found mostly a negative bias for all the seven SURFRAD sites of less than 4% when using ground-based atmospheric measurements. These last two validations do not show the performance of the models in actual operational conditions."

Comment 2. In the paragraph starting in line 44 the authors also discuss the quality of the PAR measurements and also address the error of the PAR devices. This is a bit too short given the complexity of this issue. The error of the ground measurements, when used a reference, are of outstanding importance for the model presented. There are studies already published that address the errors of PAR measurements in practice at the ground level. Schmidt et al. (2012), for instance, give an average error of 3% for PAR measurements among a large network that is, however, affected by a large standard deviation of errors observed between PAR sensors at different sites and a
reference system. A PAR sensor's response to the natural spectrum needs to be addressed briefly. Although the authors describe the quality control of the measurements used, the fact that measurements might always shows some deviance from the real value should be addressed on a certain level in a manuscript that focusses on PAR. The response of the very common PAR sensor (LICOR quantum sensor) that is also used in this study is not perfect for the natural spectrum but adheres to an optimized spectral curve. Please address this issue if the performance of your approach is measured by the correlation with those ground-measurements.

Answer: Thank you for this remark. The comment raises good points involving the uncertainty of PAR measurements. Different plants absorb at varying spectral ranges, so a PAR radiometer that would apply to all plants would be impossible to design. But, the absorption spectra for all plants are most active in the range 400 to 700 nm. That led to a proposal that PAR be given a physical definition of photon flux over 400 to 700 nm (McCree (1972). Based on this definition, an ideal instrument response would be a step function from 400 to 700 nm. The spectral response of PAR sensor used in the SURFRAD network best approximates that ideal response among commonly available PAR sensors. That is, according to the manufacturer, deviations from the ideal response, or actinity errors, based on sunlight as a source for the PAR sensors used in the SURFRAD network, are 0.3% ( $\pm$ 0.7) for most PAR sensor models used, and 0.5% ( $\pm 0.3$ ) for the latest model that began to be used in 2017. Actinity errors for other PAR sensors are on the order of 2 to 3%, with one showing an error of nearly 12%. From 1997 to 2012, Schmidt et al. (2012) compared field measurements of PAR at 84 AmeriFlux sites with a traveling, freshly calibrated standard instrument. Through 2006, AmeriFlux used the same PAR instrument as SURFRAD. For those years, the mean relative error for the field PAR instruments was -3%, but with very large standard deviation of nearly 14%. This large variability was attributed to, among other things, "infrequent calibrations, [and] inconsistent and sometimes high rates of degradation." They report that calibration drift for PAR sensors is on the order of "<2% per year although higher values (>10% per year) have been reported." The general practice

BGD
of SURFRAD operations is to replace PAR sensors each year with freshly calibrated instruments, and within the year calibration drift is monitored on a daily basis by examining the ratio of PAR to total SW broadband. If a systematic drift is detected, that PAR instrument is replaced and data during the period of drift is corrected. These two QA practices of SURFRAD eliminate the large sources of error reported by Schmidt et al. (2012) for AmeriFlux measurements.

Comment 3. The sentence in lines 75 to 78 is not quite clear, neither is the approach described. Although a reference is given, please rephrase and add a brief description about the method. How many narrow bands are used in your case? The authors mention "one or more". I would expect the model performs better the more band bins are correlated and used. How strong do the correlations have to be? The summary is too simple to understand the process. Later in section 3. 2 the method is explained in more detail, yet the statement "A total of 19 NBi is sufficient" comes out of nowhere. Please give information as to why 19 can be considered sufficient. Please also merge these two sections and explain this part of the method only once but thoroughly. Details about the origin of the function parameters (slope and intercept) are not given although it seems to be the central point of the method. How many comparisons at which stations were used to get these parameters of the affine functions? A statement like "The choice of these NBi has been made on an empirical basis." As give in lines 188 and 189 is not a sufficient description of the process. Please elaborate.

Answer: Thank you for this remark. We fully agree with you. Firstly, we have re-written this relevant part of the text and have added a brief description about the method as follows:

"The eleven KBs are KB #6 [363, 408] nm up to KB #16 [684, 704] nm. They do not cover exactly the PAR range which is limited to [400, 700] nm. In addition, the bandwidth in several KB is larger than 30 nm and may be considered large for estimating PAR in an accurate manner. A spectral resampling technique has been developed by Wandji Nyamsi et al. (2015) to overcome this difficulty. In brief, the proposed technique
gives the irradiance for every 1 nm width within each KB over the PAR range from the irradiance of 30 nm width of each KB in any atmospheric state under cloud–free conditions. The technique is explained in more detail in the section 3.2".

Then, in section 3.2, we have rewritten this relevant part of the text including all the suggestions. We would like to emphasize that no measurements have been used to develop the affine functions. They have built based on a pure modelling approach by the means of radiative transfer model.

Comment 4. Figure 3 gives the differences between the measured (at the sites) and CAMS data. While the upper panel gives a ratio, the lower panel gives the absolute difference divided by 100. It is not clear why the difference given in the original data units should be divided by 100 (unless one wants to make them appear much small than they are). Also, please provide unit labels on the vertical axes showing the differences of the various variables in the lower panel.

Answer: Thank you for this remark. We have provided figures as requested.

Technical corrections:

The sentence in line 75 starts with a dot. Please correct (delete).

Answer: Thank you. Done as requested.

In line 163 there is a dot missing at the end of the sentence. "...any location and any time."

Answer: Thank you. Done as requested.

Line167: Please spell out AFGL dataset. The acronym appears for the ïňĄrst time here and not any reader might be familiar with that dataset and its origin.

Answer: Thank you. Done as requested.

Line 137 and 138: The sentence reads":::cloud-free instants instances." Please cor-

BGD
rect.

Answer: Thank you. Done as requested.

Reference used in comments:

Schmidt A., C. Hanson, W. S. Chan, B. E. Law (2012): Empirical assessment of uncertainties of meteorological parameters and turbulent fluxes in the AmeriFlux network. Journal of Geophysical Research 117, G04014, doi:10.1029/2012JG002100.

McCree, K. J., (1972): The action spectrum, absorptance and quantum yield of photosynthesis in crop plants. Agricultural and Forest Meteorology, 9, 191-216.

---

## Author Comment (AC2) · 20 Apr 2018

First of all, we thank Referee #2 for the constructive remarks on this article. The authors believe that they have understood the concerns of the referee. All remarks have been taken into account for revising a part of the text following recommendations of the referee.

Background

In 1999 Nyamsi et al. published in Adv. Sci. Res., 12, 5-10, doi:10.5194/a descrip-

tion of an approach to derive PAR under clear sky conditions from the correlated-k approximation of Kato et al. (1999). This approach is computationallyefficient, initially applied for calculations of the broadband solar radiation under clear sky conditions in 32 specific spectral bands. The authors used this approach for assessment of the photosynthetically active radiation (PAR) from 400 to 700 nm using twelve of these spectral bands. The method has been evaluated against detailed spectral calculations of PAR derived with the radiative transfer model libRadtran. In the present study, the Copernicus Atmosphere Monitoring Service daily estimates of aerosol properties, and column contents of water vapor and ozone are used as input to the method to derive PAR under clear sky condition. The results are compared with measurements of global Photosynthetic Photon Flux Density on a horizontal plane made in cloud-free conditions at seven sites of the Surface Radiation network (SURFRAD) in the USA.

General Comments

Comment 1. The methodology used here has been described previously.

Answer: Thank for this comment. As commented by the referee #1, apparently, we were not able to emphasize clearly enough those parts of the method that have been already published from those that are discussed and published first time. We appreciate this comment and we have tried to clarify these issues in the revised manuscript. Moreover, we want to stress that the core objective was to validate this approach (at PAR range) against ground-based measurements. This is done first time in this manuscript and is entirely new contribution.

The method we described is a combination of three parts: (1) use of CAMS products to describe the atmospheric state, (2) irradiances of correlated–k approach over only eleven bands covering the PAR wavelengths by the means of libRadtran and (3) the resampling technique for computing PAR estimates. Only the third part has been previously published by ourselves. The goal of this current manuscript is to focus on the entire approach (e.g. also including the other two parts) and to present the groundbased validation.

Since estimation of PAR under cloud-free conditions at any time and place is an important first step in calculating PAR in all-sky conditions, in this paper, we concentrated first on these conditions.

Based on the above referee comment, we have re-written several parts of the text accordingly. For instance, a part of introduction is re-written as follows:

"This resampling technique has not been validated in operational conditions, i.e. using available inputs to describe the atmosphere in cloud-free conditions and the properties of the ground, and tested against ground-based measurements. This paper is making this step forward and aims at describing and evaluating the entire method when tested against measured PAR in cloud-free conditions."

Comment 2. The methodology has been implemented with actual auxiliary data to match ground observations of PAR. It was found that the bias ranges between 1-6% from the mean value. It is claimed that these errors are less than 5% than the uncertainty of the measurements. It is claimed that this demonstrates the very good level of accuracy of the proposed method (which is not obvious how).

Answer: Thank you for this remark. We fully agree with you. We have re-written the relevant parts of the text and especially we have made a discussion part, newly added in the revised manuscript.

Comment 3. Not clear what is the added value of this evaluation since the methodology itself was already evaluated. Seems, this is just an exercise what is achieved if the CAMS product is used. Would it be worse with other sources of products?

Answer: Thank you for this remark. The comment is almost similar to the comment #1. We have re-written a part of text. We have given comparisons with a method using CERES products. We found that the performances of our method are similar or better in most stations. These comparisons were mentioned in the Discussions part.

Comment 4. Not clear how this work brings us closer to obtain information on PAR under all sky conditions (information that is needed).

Answer: Thank you for this remark. We have re-written the relevant parts of the text.

Comment 5. The approach proposed is not unique and in principle, any radiative transfer model can be used to estimate PAR. Therefore, the unique contribution of the described effort was not demonstrated clearly and neither has it been shown how this gets us closer to obtain large scale information on PAR under all conditions.

Answer: Thank you for this remark. We have re-written the relevant parts of the text.

Comment 6. Relevant references are very limited.

Answer: Thank you for this remark. We have added more relevant references in the manuscript.

Comment 7. Addressing all of above concerns is needed before considering publication.

Answer: Thank you for your comments. We believe that we have understood above concerns. The remarks have been taken into account for revising a part of the text following recommendations.

---

## Author Comment (AC3) · 20 Apr 2018

First of all, we thank Referee #3 for these constructive remarks on this topic. The authors believe that they have understood the concerns of the referee. All remarks have been taken into account for revising a part of the text following recommendations of the referee.

The manuscript submitted to Biogeosciences titled, "Deriving Photosynthetically Active Radiation at ground level in cloud-free conditions from Copernicus Atmospheric Monitoring Service (CAMS) products" by Wandji Nyamsi et al. presents one aspect of a larger project developing radiation modeling for the bottom of the atmosphere. This includes a sequence of recent articles lead by the current lead author. The specific objective of this study appears to be testing the use of CAMS atmospheric products for inputs to an existing numerical radiation transfer model. The model itself is cited, as is the spectral resampling technique used here, referred to as the 'new method'. What is new in THIS study seems to be the use of the CAMS products; columnar aerosol optical depth (AOD), ozone (O3) and water vapor (H2O) inputs to the radiation model. The general accuracy of the simulated cloud-free PAR is impressively high at 7 locations with strong regional differences likely in AOD and H2O. However, a number of questions seem to remain that are central to the research objective. These include, what causes the tendency of the model to perform well at lower PAR levels and over-estimate at higher values (e.g. Fig. 2), and cause slope>1 in general? Did this appear in previous evaluations of the model? Can the differences in accuracy found between the seven stations be used to inform the cause of inaccuracy? The model over-estimate appears to be correlated with a tendency of the model to overestimate under low zenith angles and/or under low AOD. Why is this? Is the cause of this inaccuracy related to the CAMS data set or is it the radiation model itself? The answers to these questions seem important to the objectives of this study, because they should provide leads towards future model improvements. As presented, I find the manuscript needs to overcome two major obstacles currently inhibiting its potentially useful contribution to Biogeosciences.

Answer: Thank you for this comment. We feel that apparently we were not able to emphasize clearly enough those parts of the method that have been already published from those that are discussed and published first time. We appreciate this comment and we have tried to clarify these issues in the revised manuscript. Moreover, we want to stress that the core objective was to validate this approach (at PAR range) against ground-based measurements. This is done first time in this manuscript and is entirely new contribution.

[Figure]

The method we described is a combination of three parts: (1) use of CAMS products to describe the atmospheric state, (2) irradiances of correlated–k approach over only eleven bands covering the PAR wavelengths by the means of libRadtran and (3) the resampling technique for computing PAR estimates. Only the third part has been previously published by ourselves. The goal of this current manuscript is to focus on the entire approach (e.g. also including the other two parts) and to present the ground-based validation.

Since estimation of PAR under cloud-free conditions at any time and place is an important first step in calculating PAR in all-sky conditions, in this paper, we concentrated first on these conditions.

Based on the above referee comment, we have re-written several parts of the text accordingly. For instance, a part of introduction is re-written as follows:

"This resampling technique has not been validated in operational conditions, i.e. using available inputs to describe the atmosphere in cloud-free conditions and the properties of the ground, and tested against ground-based measurements. This paper is making this step forward and aims at describing and evaluating the entire method when tested against measured PAR in cloud-free conditions."

Comment 1. Surprisingly, little information is provided about the CAMS product, especially accuracy assessments of AOD, O3 and H20 vapor products from past studies. This context is necessary to assess the efficacy of using it for inputs to radiation modeling. It may also provide some useful insight into the cause of the overestimate of PAR in simulations found here.

Answer: Thank you for this comment. We have included a discussion part, newly added in the revised manuscript where we discussed about the accuracy of CAMS products as inputs of the method. We have re-written the relevant part of the text accordingly.

Comment 2. Analysis doesn't seem to test the CAMS input specially, but rather the

CAMS driven radiation simulations against observations. This could be done by comparing these simulations to those using standard (e.g. monthly average) AOD, H20 and O3 for a site. Sensitivity simulations for variations in these input variables would also help shed light on the sensitivity of model accuracy to each of these inputs (or discussed, if reported elsewhere). My guess is that this shows AOD is key. I believe AOD is also measured at the observation sites? Could the CAMS product be evaluated to determine if, for example, the overestimate could be related to a bias in CAMS AOD?

Answer: Thank for these constructive comments. The remarks have been taken into account for revising a part of the text following recommendations and suggestions especially in the Discussion part which has been newly added in the revised manuscript.

Minor comments:

Specific objective of this study is not really clear. Nor is it clearly separated from the other model developments in this sequence. There seems to be significant overlap as written.

Answer: Thank you for this remark. We have re-written the relevant part of the text accordingly.

It would be helpful to the readership of this journal for the authors to articulate more fully in the introduction and discussion sections, the implications of these methods and results to biogeoscience research in general.

Answer: Thank you for you remark. It has been done as requested in the introduction.

Methods, why not include the Modeling Efficiency Coefficient (Nash and Sutcliff 1970)? It's a very direct test of model performance, including bias and random error. There is no discussion section. This would be a useful place to investigate answers to the questions above based on the evidence presented in results.

Answer: Thank you for this valuable suggestion. The Modeling Efficiency Coefficient is very nice test mostly used to evaluate the performance of hydrological models. While

we want here to evaluate performance of a solar radiation model, the statistical indicators, namely the bias (mean of the differences), the root mean square of the differences, these same quantities but in relative values, the mean value and the correlation coefficient, the ratio and absolute differences are vastly used in many studies in the literature for such model. For this study, we think these statistical indicators are more appropriate.

The manuscript needs to be carefully edited for small but frequent lapses in grammar, or clarity in expression.

Answer: Thank you. Done as requested.

L128: What is "fine rock"?

Answer: Thank you. We have replaced these words by "small rock".

L137: End of sentence unclear-'instant instances' is confusing alliteration. Any way to rewrite using other words such as 'periods', 'frequency'...?

Answer: Thank you for this remark. We fully agree with you. We have used the word "periods".

In discussing Fig's 2 and 4, What is an 'identity line'?

Answer: The idendity line is also called line of equality or the 1:1 line. In accordance with the relevant figures, we replaced "identity line" by "1:1 line" in the text.

―――――――――――――――――――――